# Disputing the 'National Interest': The Depoliticization and Repoliticization of the Belo Monte Dam, Brazil

**Ed Atkins**

School of Geographical Sciences, University of Bristol, Bristol BS8 1TH, UK; ed.atkins@bristol.ac.uk

**Abstract:** The construction of a hydroelectric project transforms the watershed in which it is located, leading to a moment of contestation in which the scheme is challenged by opposition actors. This paper explores the interplay between pro- and anti-dam coalitions contesting the Belo Monte Dam in Brazil by discussing how each group inscribes the project with a particular resonance in policy. Drawing upon the work of Chantal Mouffe on agonism and Tania Murray Li on 'rendering technical', the subsequent discussion analyzes semi-structured interviews, questionnaires, and primary documents to explore how the storylines advanced by pro- and anti-dam actors contest the political character of Belo Monte. It is argued that within these storylines, Belo Monte's positioning within the 'national interest' represents a key site of the project's depoliticization and repoliticization—which are understood as the respective denial and illumination of the project's location within a wider terrain of political antagonism and conflict. Whilst pro-dam actors assert the apolitical character of the project by foregrounding it within depoliticized questions of economic benefits, anti-dam actors reground the project within a context of political corruption and the circumvention of dissent. With this paper providing evidence of how contests over dam construction are linked to the concealing and/or illumination of the project's political content, it is argued that the repoliticization of a project by a resistance movement can have consequences far beyond the immediate site of construction.

**Keywords:** Belo Monte; Brazil; dams; national interest; hydropower; depoliticization; repoliticization; energy policy

## 1. Introduction

Situated on the Volta Grande ('Big Bend') of the Xingu River in Pará state in the Brazilian Legal Amazon Region, Belo Monte, when completed, the Belo Monte Dam will become the fourth largest hydroelectric dam in the world. Its physical magnitude is clear—reportedly involving the excavation of over 240 million cubic meters of rock, soil, and sediment and the pouring of over three million cubic meters of concrete [1,2]. Yet, the significance of Belo Monte can also be seen in the extensive opposition staged against its planning and construction. Over 30 years, a multiactor, international coalition of nongovernmental organizations (NGOs), indigenous groups, and local communities has staged a prolonged campaign of opposition against the project. Within this 'moment of contestation' between pro- and anti-dam actors [3], both groupings have sought to locate the Belo Monte project within a wider context of both national politics and policy. This article, drawing on the analysis of public speeches, semi-structured interviews and questionnaires, and documents provided by both groupings, explores a key element of this contestation, namely the disputed positioning of Belo Monte within the notion of the 'national interest'. It is argued that these disputes over the 'national interest' represent a key site of contest over the project's relationship with the political, understood as the terrain upon which competing interests collide and contest differing visions of society. Adopting a theoretical framework informed by Chantal Mouffe's work of agonism and Tania Murray Li's

discussions of 'rendering technical', I argue that these debates are centered upon both the raising of the Belo Monte project above everyday politics and dissent and the attempted reversal of such a process. In discussing Belo Monte, proponents of the project present its construction as representing the 'national interest' in terms of the economic benefits that it is asserted to provide, with the dam perceived as necessary, urgent and beyond debate. In response to these claims, anti-dam criticism of the project seeks to reground its construction within a wider political context of vested interests and dissent, questioning the apolitical character asserted by the project's proponents by illuminating the role of corruption in its construction. In describing these storylines, this paper provides evidence of how the political—or apolitical—character of the project is rooted in a process of contestation over its location in everyday politics, antagonism, and conflict. Furthermore, it is asserted that, although Belo Monte will be built, the utility of anti-dam storylines of repoliticization extends beyond the site of this project's construction.

After detailing the methods adopted, this paper will assert the importance of pro- and anti-dam storylines in both consolidating and contesting the legitimacy of a dam project. Whilst pro-dam storylines have included a number of assertions, the analysis will focus on the positioning of Belo Monte within an asserted 'national interest'. After defining this role of the 'national interest', subsequent sections will characterize such assertions as a form of depoliticization—conceptualized in light of Mouffe's work on antagonism and Li's description of 'rendering technical'. However, it is asserted that such a process of depoliticization is reversible, with anti-dam actors able to forward alternative storylines that reconfigure pro-dam assertions and alter popular understandings of the project's construction. This paper explores such a process in relation to the construction of the Belo Monte project in Brazil. It details the interactions between pro- and anti-dam storylines and the contentious location of the project within a political terrain of corruption, economic development and the 'national interest'. Finally, conclusions are drawn.

## 2. Materials and Methods

The subsequent discussion is based on the analysis of

- 21 semi-structured interviews,
- 12 questionnaires, and
- 312 primary documents.

These materials were gathered between September 2016 and January 2018. The questionnaires and interviews analyzed are drawn primarily from local, national and international nongovernmental organizations (NGOs) publicly opposed to Belo Monte. In addition, interviews were also conducted with representatives of the Brazilian environment agency, Instituto Brasileiro do Meio Ambiente e dos Recursos Naturais Renováveis (Brazilian Institute of the Environment and Renewable Natural Resources, IBAMA), Fundação Nacional do Índio (the National Indian Foundation, FUNAI), the Ministério de Minas e Energia (Ministry of Mines and Energy, MME), and the Ministério Público Federal (Federal Public Ministry, MPF). These government agencies represent key actors in the planning, construction, and operations of dams in Brazil.

With the interviews and questionnaires primarily completed by anti-dam actors, discussions of the views, actions and storylines of pro-dam actors are based on the analysis of official speeches made by these actors. The settings for such texts included public political events, press conferences, or speeches made on the floor of the Brazilian Câmara dos Deputados (Chamber of Deputies, the lower house of the National Congress), accessible via the institution's transparency portal (http://www2.camara.leg.br/).

Additional documents included within the analysis have been disseminated by various organizations and groups, both in favor of and against Belo Monte. These include governmental sources, national and international civil society groups, domestic arms of international NGOs, and local campaigning groups. Adopting a method of discourse analysis, these materials are analyzed as communicative devices, which are written and distributed to a specific, targeted audience and for a

particular purpose. After the process of data collection and the transcription of interviews, I coded the data collected, detecting the links between the assertions of different actors and the concept of the 'national interest'. I subsequently used these links to define broader categories connected to the different storylines forwarded to either affirm or dispute the project's construction. All interviews, questionnaires, and primary source materials are referenced in the bibliography provided. (Notably, a number of the primary sources analyzed have been translated. Due to the challenges posed by such a process to the analysis of word choice and metaphor, the linguistic content of these sources is not directly analyzed. Instead, translated materials are analyzed to explore how the project is framed and located within the wider political terrain (i.e., via the development of links between Belo Monte and policies of economic development). Within this process, a variety of different storylines were detected. It is these that I outline below.

## 3. The Importance of Pro- and Anti-Dam Storylines

The construction of a dam is not an easy process—not only does earth have to be moved and concrete poured but the project must also be legitimized. With the construction of a large hydroelectric project transforming the watershed in which it is built, proponents of a dam must secure the consent or acquiescence of both local communities and wider activist networks. As a result, the planning and construction of the infrastructure is characterized by the contentious interaction between a project's proponents and opponents. Within this contest, both pro- and anti-dam actors work to locate the scheme within a wider terrain of demands and grievances to either endow it with increased legitimacy or strip such acceptability away.

The analysis of the discourses forwarded by respective pro- and anti-dam actors within this interplay provides researchers with the opportunity to analyze how both historical and contemporary hydropower projects are described in a certain way to locate them within a wider terrain of political or policy priorities. Scholarship has identified how pro-dam actors have endowed hydropower projects—and their consequences—with legitimacy by linking them to wider policies and popular demands. Examples include assertions of society's 'conquering' of nature, infrastructure as providing a techno-fix for social problems (such as water scarcity) [4,5], the relations between hydropower and statehood [6,7], and the characterization of dams as 'green', sustainable energy [8–11]. Assertions of the sustainability of contemporary dam projects provide a common storyline of legitimacy, with hydropower presented as providing clean, affordable energy that represents an alternative to fossil fuels [9,10,12,13]. However, this greening of hydropower is disputed, with recent studies highlighting the environmental impacts of the energy infrastructure—related to biodiversity loss [14,15], greenhouse gas emissions [16,17], and disrupted sediment flows [18]. In advancing these respective discourses, pro-dam actors locate a scheme within a wider social, political, or economic context—be it centered around economic growth, nationalism, or the consolidation of a certain political order. As a result, the project in question becomes inscribed with a wider political significance—endowing it with a further legitimacy.

However, unlike the infrastructure itself, the prescribed meaning of a dam is not fixed in place but remains open to reinvention and contestation. Anti-dam actors also engage in acts of discourse to present a dam project within a wider schema of resistance. Although contests over the infrastructure often encompass local concerns related to issues of the distribution of costs and benefits and the sovereignty of indigenous communities [19,20], actors opposed to a dam also incorporate additional discourses into their criticism of the project, transforming the dam projects into symbolic spaces of a wider political significance. For example, the Welsh resistance to the Tryweryn dam in the late-1950s linked the flooding of the small village of Capel Celyn to nationalist sentiments of the protection of Welsh culture and language and the subservience of the Welsh landscape to English interests [21,22]. Although unsuccessful, these assertions provided a strong counter-narrative to pro-dam storylines emphasizing the utilitarian value of the infrastructure [21].

The respective discourses forwarded by pro- and anti-dam groups can be understood as what Hajer describes as 'storylines', representing a trope or narrative that simplifies the meaning of social or physical phenomena [23]. A storyline is understood as an overarching body in which various discourses are combined into a coherent whole, with their respective complexity simplified and concealed [23]. These storylines are forwarded by discourse coalitions—defined as the collection of actors who utter a particular storyline, as well as the practices that conform to it—to legitimize positions or policies [24]. In the case of dam construction, these groupings are formed of pro- or anti-dam actors respectively, with each coalition presenting a number of storylines. For example, a pro-dam storyline may incorporate a variety of discourses including those related to nationalism, utilitarian notions of the greatest good for the greatest number and the importance of infrastructure, into a coherent storyline that is forwarded to legitimize a dam project. Although this storyline contains a range of demands and grievances, it is held together by what Hajer terms 'discursive affinity', in which previously-divergent discourses are fused by a similar way of conceptualizing the world [24].

The imposition and entrenchment of a storyline—that provides a means of understanding a project and its wider consequences—has a central role in the contentious interaction that surrounds the construction of a dam. These storylines are provided by both pro- and anti-dam actors, with each grouping locating a dam project within a wider narrative of problems, solutions, and impacts. The provision of storylines provides an important route for both groupings to inscribe a policy, process or project with a defined meaning—simplifying complex issues and debates into an accepted storyline that acts as "a catchy one-liner" that can legitimize a project, or strip such legitimacy away [23].

## 4. The 'National Interest' and Depoliticization

This article takes as its starting point one particular storyline forwarded by pro-dam actors to legitimize dam projects—that of the 'national interest'. This term—usually resonant in international diplomacy—can be understood as representing the best route forward, asserted by the government for the wider population of the state. The use of such a term implies the possibility of defining the needs and priorities of a state as a coherent, homogenous whole. At the subnational level, the 'national interest' functions as a rhetorical device designed to develop legitimacy and public support for a certain course of action [25]. These assertions of the 'national interest' can include references to the protection of the status quo (or 'security') or to national socioeconomic progress (or 'opportunity'). The securitization theory of international relations demonstrates that the description of an issue as one of 'security' acts to produce a state of emergency, invoking an existential threat [26]. A similar process occurs in the case of a storyline of 'opportunity', generating links between a policy or process and a wider popular image of national progress [27].

Previous scholarship has explored the links between dams and the consolidation of existing or emergent power structures, with the infrastructure located within a politicized context of state-building and nationalism [6,7,28–30]. For example, the Tarbela Dam in Pakistan represented a direct attempt by the central government to demonstrate and concretize its vision for a united post-independence nation [31]. Similarly, the positioning of the Rogun dam in Tajikistan within a sense of nationalist identity and patriotism allowed for the continued commitment to the project by the Tajik elite [7]. In both of these cases, the building of a dam was asserted by pro-dam actors as consolidating a new form of political rule in a fragmented state [7,31]. A similar process is currently taking place in contemporary Afghanistan; where large infrastructure projects are cast as necessary for socioeconomic development and political stability [32]. In locating hydroelectric schemes within wider policies of economic development, pro-dam actors foreground the respective projects within a wider storyline of what is good for the country—or what is in its interests, transforming the project into a central site of what is deemed the 'national interest' [8,33].

Pro-dam storylines of the 'national interest' not only function to inscribe a hydropower project with a particular meaning—casting it as a solution to certain challenges—but also act to redirect criticism of the project, restricting the possibilities of action available to anti-dam actors. This can be

seen in what Crow-Miller has labeled 'discourses of deflection' [34]. These storylines are forwarded to direct attention away from issues and outcomes that do not fit within the problems defined and solutions prescribed by pro-dam actors. For example, analyses of pro-dam storylines that present an infrastructure project as a solution to challenges of water scarcity have shown that such storylines both create a necessity of action [11,35] and restrict alternative routes forward [4,5]. Similarly, storylines of hydropower as a route to wider economic development present a vision of the benefits provided by the project, foregrounding it within a wider national interest of progress and opportunity [7]. For example, pro-dam actors supporting the construction of the Volta River Project in Ghana presented the population displacement caused by the dam as an opportunity, with the displaced sacrificing their "traditional homes in the interests of the nation" [36]. Such a storyline of opportunity—framed in relation to the asserted 'national interest'—deflected criticism of the project, based on population displacement, by foregrounding such impacts within a context of national progress, allowing for a further legitimacy of action.

Storylines of the national interest provide a route to control the policy agenda, allowing for the deflection of criticism, the restriction of alternatives and the consolidation of existing asymmetries of power [34,35,37]. Pro-dam actors forward storylines to not only frame a project in a particular light or related to a particular issue but to also discredit opposition networks, representing the discursive marginalization of alternative voices resisting the project in question [38,39]. For example, in the contest over the Sardar Sarovar dam in India's Narmada Valley, opposition networks were marginalized and trivialized by pro-dam actors' claims of the 'national interest', with activists labeled as 'youngsters', 'boys and girls', and 'eco-fundamentalists' to position them outside of the asserted interests of the Indian state and discredit their credentials as political opposition [40]. In making these statements, pro-dam actors forwarded storylines to delegitimize opposition actors, casting their grievances and demands as unacceptable or subversive. This transforms the contestation surrounding the project from a political struggle over a contested future to a simple binary of 'for/against', 'patriotic/treasonous', and 'good/evil' [41], with pro-dam actors locating opposition groups and alternative viewpoints as existing both beyond the 'national interest' and standing in the way of future progress.

In this light, pro-dam storylines both assert the importance of a project and delegitimize opposition networks by rendering a project or policy process as apolitical and separating legitimate from illegitimate actors, demands and grievances [41]. Tania Murray Li's concept of 'rendering technical' provides an important route to understanding how pro-dam storylines of the 'national interest' constitute this normative divide between the necessity of a hydroelectric dam and the illegitimate demands of anti-dam actors. Li argues that proponents of development projects and policies often present these plans in technical terms, restricting their political content. In doing so, the political character of a project is stripped away, with policy problems and their associated solutions defined as necessary, urgent, and apolitical [42]. As Li has argued "questions that are rendered technical are simultaneously rendered nonpolitical . . . (to) exclude the structure of political–economic relations from their diagnoses and prescriptions" [42]. It is this 'rendering technical' that removes projects from the terrain of political contestation—delegitimizing the dissent of those groups standing against their construction.

A pro-dam storyline of the 'national interest' provides a simplifying lens that presents dam projects as a technical and apolitical by configuring the normative divide between legitimate and illegitimate thinking, national progress and regressive activity and isolating the opposition network from the wider community, with anti-dam actors cast as existing outside or what is deemed 'the national interest'. This act of exclusion constitutes the depoliticization of the hydropower project, in which "legitimate and responsible actors' demands are distinguished from illegitimate, irresponsible actors and unrealistic and impossible demands" [41]. In advancing these depoliticizing storylines, powerful actors assert the need for agreement and consensus whilst concealing alternative interpretations and visions of a project's importance, as well as the political interests and identities that both argue for and benefit from its construction [43].

In applying this concept of depoliticization, I draw from the distinction between 'the political' and 'politics' proposed by Chantal Mouffe [44]. Within this reading, 'politics' refers to the numerous practices, institutions and acts of discourse that establish a certain order and organize and manage society. 'The political' refers to the occurrence of antagonism that is present in all society, with different social groups and interests competing to achieve the partial dominance of their own worldview within these practices and institutions [44,45]. This antagonism does not necessarily have a negative character but can instead be understood as essential to democratic politics [44–47]. A well-functioning democracy often requires a clash between adversaries possessing political positions that are recognized as legitimate—it is this recognition that allows for the transformation of antagonism into agonism, with opponents recognizing their adversarial relations but not as enemies [47]. For Mouffe, it is the persistent occurrence of this agonism that allows for the airing of political conflicts, social demands, and ideological contests of what society should be, or what is in a society's interests [44,46–48]. As Mouffe has argued, "political questions are not mere technical issues to be solved by experts. Proper political questions always involve decisions that require making a choice between conflicting alternatives" [46]. Within this reading, political disagreement is to be welcomed as it allows for the airing of competing grievances and demands. Yet, it is this agonistic character of the political that depoliticization limits, denying the legitimacy of alternative positions and storylines and the political character of dam projects. Such assertions can also be found in the work of Swyngedouw—drawing on the work of Jacques Rancière —on the post-politics of environmental policy and climate change, with such challenges often asserted as "to be dealt with through compromise, managerial and technical arrangement, and the production of consensus" [49,50].

However, this process of 'rendering technical' is reversible. As Li argues, communities and opposition groups—standing against development projects—can forward arguments of critique to puncture a technical discourse, providing it with "a challenge it cannot contain" and opening up a new front of struggle [42]. Political movements and outbursts of dissent act to forward and shape alternative understandings and trajectories [43], engaging in an act of reconfiguration that simultaneously challenges and modifies dominant pro-dam storylines. Groups and individuals opposed to a dam forward their own storylines of resistance to not only critique the project itself by illuminating its social and environmental impacts but to also challenge the pro-dam storylines of legitimacy. Whilst scholarship has explored how anti-dam actors contest the planning and construction of dams by forwarding storylines that illuminate the social and environmental impacts of respective schemes [20,51–53], it is important to interrogate how these resistance storylines simultaneously challenge and reconfigure dominant pro-dam narratives.

Whilst the state-based character of international relations may allow for the invoking of a unified interest of a state in diplomatic terms, the extent of differences within such a state challenge the potential entrenchment of a storyline of the national interest at the subnational level. The heterogeneity of society results in the national interest becoming subject to multiple definitions and constructions [54]. Whilst pro-dam actors may seek to legitimize a hydroelectric project by describing it as in the national interest, anti-dam actors may challenge such an assertion by illuminating the divergence of the project from the wider interests of the population. As a result, how a dam project is defined in relation to the 'national interest' becomes a key site on which the contest between pro- and anti-dam actors takes place.

The national interest is not a unified, agreed-upon concept and is, instead, subject to multiple, contesting definitions—allowing for anti-dam actors to appeal to additional understandings and storylines as a means to generate opposition against a dam project by challenging its location in the national interest. Anti-dam groups also forward storylines that illuminate the political meaning of the dam, focusing attention on the political interests pursuing its construction, highlighting disconnects between those impacted by a project and end-users of the energy generated and emphasizing calls for justice for those local communities affected [20,55,56]. This is evident in the actions of the Narmada Bachao Andolan (NBA), opposed to the Narmada dams in India, which engaged in a reconfiguration

of the pro-dam storyline of development that accompanied the project to illuminate its equivalence to the destruction of cultural heritage and the maldistribution of costs and benefits [40]. This alternative vision involved the reversal of a process of depoliticization to provide an alternative vision of the dam project in question and the interests underpinning its construction. I term this process 'repoliticization', understood as the adoption of storylines to reveal the competing interests that underpin environmental projects and articulating these within an alternative vision of society [41,57]. This process of rendering visible the political character of a project is in response to previous claims of 'national interest', resulting in the labeling of this process as 'repoliticization', rather than 'politicization'—which is understood as the activation of a political element of a project or policy that has not been previously denied. This process of repoliticization represents the generation of debate over alternative futures, reactivating the politico-ideological character of hydraulic infrastructure that is muted by storylines of depoliticization.

Although the opposition to the Narmada projects was ultimately unsuccessful—with the Sardar Sarovar dam officially opened in September 2017—the movement against it provides an important example of how anti-dam groups reconfigure dominant pro-dam storylines and alter the trajectory of hydropower projects elsewhere. The storyline forwarded to illuminate how hydropower projects impacted the local Adivasi community allowed for the growth of the anti-Narmada opposition network, with the movement formed of numerous interests at all levels that discussed the benefits and impacts of large development projects on a more general basis [58]. The advancing of this anti-dam storyline engaged in a direct critique of the notion of economic development forwarded to legitimize the dam, pressuring the World Bank—a key funder of hydropower projects—to establish an independent review to assess the impacts of the project. The significance of this process is evident in how the opposition to the Narmada project extended beyond the provincial and localized activism and came to represent a wider outpouring of dissent against hydropower. This is evident in how the NBA contributed to the World Commission on Dams 2000 report, with the impacts of the project providing evidence of the lack of cultural heritage studies and mitigation measures within 20th-century dam building [59]. In doing so, those opposed to the Narmada projects demonstrate how activism can repoliticize hydroelectric projects, elevating localized impacts into a terrain of national and international significance.

This study will explore how the links between hydropower and the 'national interest'—as asserted in pro-dam storylines—are not set in stone but are, instead, contested by anti-dam actors. In doing so, it explores how the 'national interest' provides a key site of the interplay between pro- and anti-dam groups contesting Belo Monte. Whilst pro-dam actors forward a storyline of the national interest to strip a hydropower project of its political meaning and implications, anti-dam groups seek to illuminate the context and controversy of its construction, as well the political interests pushing the project forward.

## 5. Belo Monte

The Belo Monte dam—due for completion in 2019—is projected to generate a maximum of 11,233 megawatts, to be distributed via high-capacity transmission lines over 2000 km to the southeast of Brazil, where the majority of national energy demand is based [60,61]. With the roots of its development found in the era of military dictatorship that ruled Brazil from 1964 to 1985, the planning and construction of the project has involved a prolonged, 30-year period of contestation between its proponents and opponents. A previous incarnation of the project—then named Kararaô—was met with an extensive opposition campaign, which forwarded a storyline of resistance based on the rights and territory of indigenous communities and environmental health [8,51]. This opposition culminated in the 1989 Altamira Gathering, with local indigenous communities forming a formidable anti-dam coalition with national and international organizations and celebrities to protest the project. In response to this opposition, Kararaô was removed from national energy plans in 1990. However, the project returned in the early-2000's as a key part of the national development policy agendas proposed by the government of the Partido dos Trabalhadores (Workers' Party, PT), led by Presidents

Luiz Inácio Lula da Silva (2003–2011) (herein Lula) and Dilma Rousseff (2011–2016) (herein Dilma). A new scheme—now named Belo Monte—was formally proposed by the government in 2005, receiving funding from the Brazilian development bank—Banco Nacional de Desenvolvimento Econômico e Social (National Bank for Economic and Social Development, BNDES). Additional supporters of Belo Monte include actors from the Ministério de Minas e Energia (Ministry of Mines and Energy, MME), regional and national energy companies (including the national utility Eletrobras and, the owner of Belo Monte, Norte Energia) and companies tasked with the project's construction. Lining up against Belo Monte was a multilevel coalition, including indigenous groups and local communities, regional and national nongovernmental organizations (NGOs), and international campaigning organization (such as Amazon Watch, International Rivers, and Greenpeace).

The contestation surrounding the Belo Monte dam provides a productive case for understanding the role of the political in the management of water and the infrastructure that governs it. The project is a constituent part of a wider developmentalist agenda, being funded as part of the Programa de Aceleração do Crescimento (Growth Acceleration Program, PAC)—a policy package representing significant spending on infrastructure in Brazil. At the time of its planning, it was one of at least 25 large hydroelectric projects planned to have been built in Brazil's Legal Amazon region [61]. As a result, it is tied to additional issues and priorities present on the political landscape, including economic development and energy security [8,51]. Whilst previous research has uncovered numerous pro-dam storylines forwarded to legitimize the Belo Monte project, including assertions of the project's sustainability and its role as a solution to issues of energy insecurity [8,12,62], this study focuses on a particular storyline—that of the national interest. It is argued that this storyline provides a significant site of contestation—with the role of the project in the political subject to dispute. Whilst pro-dam actors have forwarded storylines that depoliticize the project, anti-dam actors have sought to render visible the political connotations of Belo Monte by regrounding them within a wider context—distant from assertions of the 'national interest'. In doing so, these competing groupings provide a lens through which explore how the political elements of a water infrastructure project are not set in stone but are, instead open to contestation.

## 6. Pro-Dam Assertions of the 'National Interest'

Within the pro-dam storylines, the positioning of Belo Monte within 'national interest' is rooted in the political economy of the hydroelectricity to be generated by the project. Pro-dam actors in Brazil present the importance of Belo Monte as rooted in the economic benefits that it will stimulate, both for the local region and beyond. For example, in 2011, President Dilma Rousseff defined the scheme as a "fundamental undertaking for the development of the region and the country" [63]. In the same year, the then-Senator of Rio de Janeiro (2007–2014), Francisco Dornelles (Partido Progressista, PP), argued that "the construction of Belo Monte dam is of (the) greatest importance for the development of the country to sustain economic growth, (and) job creation" [64]. In making these statements, the pro-dam actors assert the links between the Belo Monte project and the economic future of the region, providing benefits for a diverse group of beneficiaries [8]. The Federal Deputy for Amazonas, Carlos Souza (Partido Progressista, PP) argued Brazil "need(s) the Belo Monte hydroelectric project so that this country can continue to grow" [65]. Within this storyline, Belo Monte represents a symbol of a shared future of economic development and an integral part of the Brazilian national interest—with the project presented as the only route forward [66].

In asserting the economic benefits to be stimulated by the Belo Monte project, pro-dam actors present the projects as apolitical, technical projects that are in the 'national interest' [8,33]. For example, José Carlos Aleluia, a former president (1987–1989) of the Companhia Hidrelétrica do São Francisco (São Francisco Hydroelectric Company, CHESF) and Federal Deputy for Bahia (Partido da Frente Liberal, PFL), labeled the project as "not a government project, (but) a project of the nation" [67]. In making this statement, Aleluia articulates equivalence between the Belo Monte project and the 'national interest', affirming that the dam represents a national effort towards energy security that

would provide benefits for the population across Brazil [67]. These allusions to a shared common future were repeated by a number of other pro-dam actors when discussing the Belo Monte project, with the project affirmed as providing shared benefits, widening the perceived beneficiaries of the energy provided [63,64,68].

This storyline of the national interest is forwarded by pro-dam actors to raise Belo Monte above the political, elevating it beyond the everyday politics of agonism and delegitimizing opposition actors. By presenting an equivalence between Belo Monte and a shared future, pro-dam actors, such as Aleuia, draw a division between those in support of the scheme (sharing in the common future) and those opposed to the project, existing both outside the defined 'national interest' and standing in its way. Anti-dam actors were described as holding Brazil back from the economic development promised by the Belo Monte dam. For example, Asdrubal Bentes, a Federal Deputy (Partido do Movimento Democrático Brasileiro, PMDB), argued in 2001 that potential resistance to the Belo Monte project represented an "intolerance from those who do not want to see this country develop (and to be) producing well-being for its people" [69]. In asserting such 'intolerance', Bentes distinguishes a divide between 'legitimate' and 'illegitimate' grievances and demands. The arguments of opposition groups become representative of the demands of those who, in the words of Bentes in a later speech "wish to derail our development, who want to stifle the Amazon and not allow us to make the most of our hydroelectric potential, to use our riches in favor of Brazilians" [70]. It is by forwarding such storylines that pro-dam actors cast opposition actors as existing outside of the legitimate order and exclude them from democratic debate, stripping the contest surrounding Belo Monte of its contentious character by marginalizing those campaigning against it.

A storyline of depoliticization is also evident in a 2010 speech made by President Lula in Altamira, the city nearest to Belo Monte's construction site. Within this address, Lula compared the opposition coalition to his personal experience protesting the Itaipu dam, arguing that his opposition was the result of a lack of information and awareness of the importance of such projects. He argued that:

> "The opposition (to Itaipu)—like these kids (those opposing Belo Monte)—for lack of information, used to say that an earthquake would happen, say that the Itaipu reservoir would cause an earthquake in the Itaipu region (and that) the weight of the water would change the Earth's axis".  [71]

In making this comparison, Lula casts the opposition to Belo Monte as 'kids' who are naive and mistaken, highlighting what he perceives as their having a limited knowledge of the project and the problems that it is intended to solve [71]. Drawing on his own experience opposing Itaipu, Lula highlights that the contemporary opposition to Belo Monte will, with time, understand the importance of the project presenting an image of anti-dam actors as not only blocking the fulfillment of Brazil's national interest but doing so based on incomplete information and illegitimate grievances [8].

Within this pro-dam storyline of depoliticization, anti-dam groups are presented as voicing unfounded, reactionary, and illegitimate demands that should not hinder the construction of Belo Monte. As Francisco Dornelles argued in 2011, Brazil cannot "allow partial views of reality to prevail in the face of general interest" [64]. The storyline depoliticizes the contest surrounding Belo Monte, rendering its construction as a technical pursuit devoid of political content, with the opposition actors characterized as misguided, ignorant, and restricting Brazilian economic development.

*Rendering Belo Monte Technical*

A process of depoliticization is also evident in the pro-dam coalition's use of legal mechanisms to continue planning and construction of Belo Monte. Throughout the project's planning and construction, numerous legal challenges have attempted to secure the suspension or cancellations of the Belo Monte project. The Brazil state prosecutor, the Ministério Público Federal (MPF) referred 25 lawsuits against the Belo Monte project between 2001 and 2016—with these legal arguments voicing the grievances of local communities regarding a failure of Norte Energia to consult with and secure the consent of

local populations, the methodological flaws present in official Environmental Impact Assessments, and the challenges faced by Norte Energia in meeting a number of conditions for its license to build the project [72]. However, these legal challenges have been ultimately unsuccessful, with the construction of Belo Monte continuing.

A key factor in this persistence of the project can be found in the use of a legal mechanism that allows the judiciary to overthrow previously-made legal decisions in the name of the 'national interest'. This instrument—known as the Suspensões de Segurança ('Security Suspension')—dates back to the years of the military dictatorship and Law 4348/1964, which allowed for the suspension of judicial decisions—based on the criteria of if the act/decision could cause "serious damage to the health, safety, order, and public economy." The use of a Security Suspension allows for the overruling of court decisions—often based on securing individual and collective rights—to ensure the continuation of policies or completion of projects that are defined as being in the 'national interest'. The mechanism allows for a judge at a higher court (be it on the Federal Circuit or at the Brazilian Supreme Court) to overturn a decision made by a judge on a lower, more-local circuit. Such a decision to overturn must be based on the belief that the initial decision risks challenging the "health, safety, order, and public economy" of the Brazilian state and simultaneously restricts the right of appeal, allowing the project to continue without interruption [2].

In the case of Belo Monte, Security Suspensions have been used to overturn previous decisions that have called for the suspension of the project's construction [73]. With this judicial mechanism predicated on linking a decision to a defined 'national interest', the security suspensions represent a judicial resource appealed to by pro-dam actors to overturn legal decisions that suspend, delay or cancel Belo Monte's construction. The use of such a mechanism affirms the exceptionality of the project in question, suspending the legal rights of those affected whilst asserting the significant urgency of the project's completion. In doing so, the use of a Security Suspension not only provides a mechanism of legal intervention but also a significant means to redefine its position within the legal space itself, raising the project above both legal procedure (as evident in its restriction of the right of appeal) and the political (in its declaration of a project as being in the 'national interest') [73]. The importance of Belo Monte becomes located at the national level, with local communities impacted by the project unable to secure judicial redress against Norte Energia or the pro-dam actors calling for the dam's construction [73].

The use of Security Suspensions demonstrates the use of judicial resources to assert and reaffirm the exceptionality and significance of Belo Monte, whilst denying its political character. By rooting the construction of Belo Monte in the 'national interest', the use of Security Suspensions act to depoliticize the project, raising it above everyday judicial procedure. This is illustrative of Li's 'rendering technical', with alternative viewpoints—related to social and economic impacts or a failure to consult with local populations—cast aside and the hydroelectricity to be generated by Belo Monte presented as both urgent and necessary. This functions to strip the project of its political character, raising it above the terrain of the political and delegitimizing opposition. By defining Belo Monte as an integral part of the 'national interest', Brazilian judicial actors not only articulate the equivalence between a project and economic development but also exclude other elements (related to impacts or noneconomic costs and benefits, for example) to present the respective projects as technical, rather than political, decisions.

## 7. Illuminating the Political

With the pro-dam coalition forwarding storylines that foreground the construction of Belo Monte within the 'national interest', anti-dam groups challenge this equivalence by illuminating the political context in which Belo Monte has been planned and built. In discussing the project, anti-dam actors made direct reference to the assertions of the 'national interest' contained within the legitimizing storylines forwarded by the project's supporters. However, rather than merely challenging declarations of necessity and urgency, anti-dam actors at the regional, national, and international level sought to reconfigure the ties between Belo Monte and what is deemed to be in the 'national

interest'. In doing so, anti-dam actors rendered visible a wider context and controversy of the project's planning and construction, regrounding their opposition within criticism of political corruption and the circumvention of democracy. This represents a process of repoliticization, with anti-dam actors illuminating the political background and implications of Belo Monte to cast doubt on the project's role within the wider national interest.

When challenging the importance of Belo Monte to the 'national interest', anti-dam actors highlighted the personal and political commitment of a number of individual politicians—as well as the wider government of the Partido dos Trabalhadores (PT) [74–79]. Within this alternative storyline, the project was presented as equivalent to—and symbolic of—the wider policies and principles of the government of the PT and Presidents Lula Dilma Rousseff, who successively governed Brazil between 2003 and 2016 [74,75,77,80,81]. In forwarding such a storyline, anti-dam actors challenged the pro-dam storyline of the 'national interest', arguing that the projects, instead, represent the pursuit of the political and economic goals of a particular political group [74,75,77,82]. Within this anti-dam storyline, Belo Monte is presented as "the head carnival float for the party. The showcase to show that the party, the President (is) helping Brazil develop and so forth" [83]. As one interviewee based at the Brazilian arm of an international EO, argued, the announcement of hydropower projects coincided with election years:

> "When Lula wanted to (be) re-elect(ed), he launched the Madeira dams. When Dilma came, she launched Belo Monte. When they wanted the second term of Dilma, they built São Luiz do Tapajós. It was so weird. In twelve years, they had three big dams to launch exactly in the election year(s)". [75]

In making this statement, the interviewee highlights the links between the construction of a number of hydropower projects—including the Madeira river dams (completed in 2012) and the São Luiz do Tapajós (removed from energy plans in 2016)—and the electoral goals of the PT. This asserted link repoliticizes the respective projects by foregrounding them within the political, represented by the agonism of partisan politics and political elections. According to the anti-dam actor quoted above, the pro-dam storyline of economic benefits is asserted to generate popular support for the government building them, allowing continued electoral success [75].

In light of this deemed political and electoral importance of hydropower projects in Brazil, opposition groups highlighted the links between Belo Monte and the ambitions of the PT. Anti-dam actors defined the economic doctrine of the PT government as representative of a policy of 'development at all costs', with the aim of economic development pursued with limited concern for social or environmental impacts or dissent from local populations [74,75,78,79,81,84]. By emphasizing these links, anti-dam actors highlight how members of the PT government exerted political influence to ensure the realization of the Belo Monte project, regardless of its social and environmental impacts [74,77,85–87]. For example, anti-dam materials argued that political pressure was applied to technical staff in IBAMA, resulting in the construction of Belo Monte without adequate mitigative measures to address the impacts of construction [86–88]. In 2009, two senior officials at IBAMA—Sebastião Custódio Pires and Leozildo Tabajara da Silva Benjamin—resigned from their roles in the organization, complaining of high levels of political pressure to approve the Belo Monte project [88]. Two years later, IBAMA President Abelardo Bayma Azevedo also resigned for similar reasons [87]. Similarly, the prominent anti-dam activist Telma Monteiro accused Lula of ignoring information on Belo Monte's environmental impacts that have been provided by scientists and researchers [89]. An interviewee highlighted that members of the PT government had restricted the publication of a report that demonstrates how Belo Monte will be affected by the flow of the Xingu being reduced by future climate change [83]. In emphasizing these examples, anti-dam actors assert the political agency of pro-dam actors, with the proponents of Belo Monte asserted to be driving the projects forward with little attention paid to dissenting views.

A prominent example provided by anti-dam actors to highlight the pro-dam coalition's commitment to the Belo Monte project concerns the Brazilian response to the 2011 decision of the Inter-American Commission on Human Rights (IACHR) to request the suspension of Belo Monte's environmental licensing process, due to its effect on the indigenous populations of the region [74,77,90]. In response to this decision, the Brazilian government withdrew its candidate for a seat on the commission, withheld its annual payment from the Organization of American States and threatened to leave the organization entirely [91]. The IACHR rescinded the decision soon after. This episode is presented by opposition actors as indicative of the undemocratic context in which the projects were developed, with limited room for dissent and judicial appeal [74,80,92]. For example, one interviewee explained, "Belo Monte was a very important government decision. The government said "Belo Monte is a decision" and (this) never changed" [80]. In making this statement, the anti-dam actor illuminates the political commitment to the Belo Monte project, arguing that pro-dam actors pursued the project unilaterally, with limited opportunity for alternative voices.

When discussing this political commitment to the projects studied, numerous opposition materials referred to the words of Lula made at a 2009 meeting, in which he promised that "Belo Monte will not be shoved down anyone's throat" [93,94]. This statement has become a central narrative device in the resistance to the Belo Monte project, with anti-dam actors highlighting Lula's failure to keep his promise, despite the complaints of local communities [90]. For example, a letter to President Lula in 2010, organized by Amazon Watch and signed by numerous international NGOs, argued that:

> " . . . Regardless of these concerns from your fellow Brazilians and your earlier promises to them, we see that your government indeed intends to shove Belo Monte down the throats of the directly affected Indigenous and riverine communities in the Amazon". [82]

Within this statement, Lula's words become rearticulated to highlight how commitment to the Belo Monte projects by the governments of Lula and Dilma is representative of a 'steamroller' has neglected the social and environmental impacts of the project [81,83,93,95,96]. As one interviewee, an international journalist who had reported on Belo Monte, described "They basically rammed Belo Monte through, despite vast opposition within Brazil itself and from the international community" [97]. In arguing that the actions of pro-dam actors have restricted the agency of those opposed to Belo Monte, anti-dam actors challenge the use of Security Suspensions in the construction of the Belo Monte dam [76,81,83,90]. The use of these legal instruments to dismiss legal opposition is presented as evidence of the circumvention of democratic practice by a corrupt pro-dam coalition intent on building hydroelectric dams in the Brazilian Amazon [84]. For example, a respondent based at a Brazilian human rights organization labeled such judicial verdicts as a "political decision that neutralizes justice" that show "the limit of the democratic institutions (of Brazil)" [90]. Similarly, the Security Suspensions were presented by a representative of a nongovernmental organization that has written extensively on these mechanisms as "the manipulation of the justice system to legitimate a project of dubious legality" [84]. In making these statements, anti-dam actors present Belo Monte as representing the circumvention of democratic and legal norms and institutions in Brazil, with the judiciary reduced to a rubber-stamp of legitimacy for the pro-dam coalition and the opposition to Belo Monte having limited opportunity to reverse these decisions [72,74,96].

*A 'Promiscuous' Relationship*

The assertions of the importance of political interests in ensuring Belo Monte's construction was further developed by anti-dam actors to illuminate the alleged links between the dam and a wider corruption scandal that has dominated Brazilian politics in the years since 2014. This was in response to the high-profile Lava Jato ('Car Wash') investigation that exposed a culture of corruption at the center of Brazilian politics. This investigation started in 2014 as a probe into money laundering at the Posto da Torre (Tower Gas Station) in Brasília but soon widened to become an expansive anticorruption investigation into an intricate web of political and commercial corruption. The enquiry

uncovered an extensive scheme of corruption centered on the semi-public oil company, Petróleo Brasileiro S.A. (Petrobras), where executives were allegedly paid bribes to award contracts to favored construction companies. This money would be funneled to politicians, funding election campaigns that kept the governing coalition in power. The resultant investigation ensnared a number of Brazilian construction companies involved in the Consórcio Construtor Belo Monte (Belo Monte Construction Consortium, CCBM), including Odebrecht, Camargo Corrêa, and Andrade Gutierrez, as well as the political parties in government during the years of Lula and Dilma's presidencies, such as the PT, the Partido Progressista (Progressive Party, PP), and the Partido do Movimento Democrático Brasileiro (Brazilian Democratic Movement Party, PMDB).

In discussing Belo Monte, numerous anti-dam actors foregrounded the planning and construction of the dam within this wider context of corruption uncovered by the Lava Jato investigation [74,98,99]. These actors adopted a number of terms to describe the wider pro-dam coalition building Belo Monte. For example, a letter from anti-dam actors, congratulating Dilma on her election as President, argued that Belo Monte was being pushed forward by what is described as the 'relações promíscuas' ('promiscuous relationship') between political and commercial actors [100]. Although making no direct reference to the occurrence of corruption that was to be uncovered, by referencing an act of 'promiscuity' this letter—written in 2011—presents the nature of this relationship as both close and immoral [100]. Similarly, writing for International Rivers, Zachary Hurwitz coined the term 'hydro-mafia' [85]. This evokes a criminality in the actions of proponents of Belo Monte, with Hurwitz's use of the term alluding to the violent, covert and ruthless character of the pro-dam lobby.

Anti-dam actors presented this 'promiscuous relationship' as an explanatory factor for the construction of the Belo Monte project—with the dam presenting an opportunity for the alignment of the interests of the two groups and, with it, corruption [74,75,77,100]. For example, a representative at an international environmental organization (EO) described the project as a "situation that was ripe for corruption to reign, to flourish" [77]. The same interviewee later explained "this project was perhaps not only a source of corruption, it was perhaps built because of corruption. It was justified in their minds by the fact that vast quantities of public funds would fall into their coffers" [77]. Furthermore, an interviewee from an international EO argued that Belo Monte represented a corrupt exercise, designed to benefit a limited few: "the two ruling political parties, the PT and the PMDB were essentially splitting the tips, they're splitting the corruption benefits, revenues from the companies who were getting these enormous contracts in thanks for their having run the project forward" [77]. This statement positions the Belo Monte project within the political context of the corrupt practices uncovered by the Lava Jato investigation to assert that it is this corruption that led to the dam's construction. Furthermore, an article published by the anti-dam NGO, International Rivers argued:

> "As the investigations of Operation Lava Jato have revealed massive corruption within the Brazilian dam industry, the fundamental reasons for the federal government's obsession with destructive dam projects particularly during the administrations of Luis Inácio Lula da Silva and Dilma Rousseff—are becoming increasingly clear". [101]

In light of this perceived presence of corruption, anti-dam actors presented the use of Security Suspensions as "a scam in order to benefit the public power (defined as the Brazilian government), in alliance with the companies behind big projects" [79]. This statement highlights the belief of many in the opposition that these mechanisms both allowed and legitimized the corrupt relationship between Brazilian political actors and the construction sector evident in the Belo Monte project [77,79,83]. In presenting the links between the use of Security Suspensions and the Lava Jato investigation, anti-dam actors expand the alternative vision of the reasons behind the construction of the Belo Monte project, characterizing it as for the benefit of the few, rather than for the economic development of Brazil, as asserted in pro-dam storylines. As one interviewee at an international environmental organization argued, the use of Security Suspensions

> "Smacked as a pretense to run the project forward to the benefit of the companies who are building it, who are not investing in it. They are simply the ones earning the massive contracts and also the politicians who were making sure that this project (Belo Monte) moves forward by hook or by crook". [102]

Within this statement, the interviewee challenges the assertion of the 'national interest' present within the pro-dam storyline by highlighting the links between Belo Monte and the Lava Jato investigation. This functions to directly dismiss the arguments of 'national interest' that are present within the judicial decisions that suspended previous judgements against the Belo Monte project [77]. It is within the storyline of repoliticization that, as one interview explained, Belo Monte becomes presented "part of a political project that has the elite as the main actor that would benefit from these choices" [74]. Anti-dam actors argued that it is these benefits, shared between political and private actors that resulted in the construction of the project, regardless of the grievances and demands voiced by local communities and opposition groups [77,100–104]. For example, domestic resistance actors published a letter to Dilma Rousseff that argued that the closeness of this relationship has resulted in pro-dam actors neglecting the grievances and demands of the local communities impacted by the project [95].

In emphasizing the links between Belo Monte and corruption, anti-dam actors argue that the project's construction was not in the name of 'the national interest' but, instead represented the narrow interests of political and commercial actors engaged in corrupt behavior. The presentation of the links between Belo Monte and Lava Jato received increased coverage in both national and international media reporting of the Belo Monte project between 2015 and the time of writing [105–108]. Anti-dam actors argue that although they had reported on the potential of corruption in the Belo Monte project, it was not until the exposure of the scandal by the Lava Jato investigation that these concerns resonated with the wider Brazilian population. One interviewee based at a domestic human rights NGO explained, "We have been calling everybody's attention to that, but nobody heard that until the Lava Jato came and started to find out crimes and corruption" [74]. The exposure of this corruption constitutes a key site in the resistance against Belo Monte, allowing it to develop a wider resonance in a Brazilian society already protesting corruption. As the prominent journalist Leonardo Sakamoto argued, it was the exposure of this corruption that finally turned popular opinion against the Belo Monte project—with its role in the national interest discredited [109].

This analysis has demonstrated that the storylines forwarded by these resistance actors reconfigure dominant pro-dam storylines of the national interest by illuminating the relationship between the project and political interests and actors. Such a process of reconfiguration elevates local anti-dam actors to the terrain of antagonism that constitutes the political by outlining an alternative reading of hydraulic infrastructure and the drivers behind their construction. In doing so, the trajectory of contestation between pro- and anti-dam actors in Brazil is altered, with reconfigured storylines of the national interest weakened in their popular appeal. Rather than a technical, apolitical project in the 'national interest'—as described in the pro-dam storyline—Belo Monte is presented as both equivalent to and a result of the corruption scandal that dominated Brazilian politics at the time of its construction. A Federal Deputy interviewed, who has often been vocally critical of Belo Monte, argued that it was this corruption that led to a context in which "the environmental laws, human rights standards were being bulldozed to expedite this project" [102]. The economic benefits promised by pro-dam actors are rearticulated to represent a political commitment to the planning and construction of hydropower projects, regardless of their social and environmental consequences or the grievances of the local population. The illumination of the links between the projects studied and the commitment of Lula and Dilma highlights the significant political interests behind the dams and regrounds the resistance to them within the Mouffean political of competing interests and worldviews.

Although the Belo Monte project has been built and entered operations, this altered trajectory has consequences beyond the immediate site of its construction. This is evident in the suspension of an additional dam project in the Brazilian Legal Amazon Region. In 2016, the São Luiz do Tapajós

project—an 8040 MW hydroelectric dam—was removed from the environmental licensing process and energy plans. Whilst official sources asserted that this decision was in response to an uncertainty regarding the project's social and environmental impacts [110,111], anti-dam actors argued that the Lava Jato investigation, its prominence in national new cycles and the links between this corruption and Belo Monte provided a brake on future hydropower projects, such as the São Luiz do Tapajós scheme [77,78,98,99]. The logic underpinning this argument is that the exposure of the corruption related to the Belo Monte project led to the loss of the political capital held by the pro-dam coalition [112,113]. For example, a representative of a domestic EO argued that the Lava Jato scandal had left the pro-dam coalition on the back foot:

> "They (the pro-dam coalition) are unstructured, without money (and) in prison. There's no one. There's no money. Everything is broken. I think that's what explains the cancellation of the São Luiz do Tapajós (dam) is much more a political and economic situation than the environmental evaluation (IBAMA's decision) suggests". [80]

In making this statement, the interviewee asserts that the exposure of the corrupt relationship between politicians and commercial actors by the Lava Jato investigation has restricted the political commitment to hydroelectric projects—with the exposure of what Zachary Hurwitz described as a 'hydro-mafia' restricting the political capital available to be invested in such projects. For the interviewee above, the corruption scandal had exposed the impunity of the companies implicated, restricting their ability to be involved in large infrastructure projects in the future [80]. Although Belo Monte has been built, the significance of the anti-dam storylines forwarded to dispute its construction have a wide significance. Anti-dam actors locate the construction of Belo Monte and suspension of the São Luiz do Tapajós dam within a shared context, with the removal of the São Luiz do Tapajós project from the environmental licensing process traced to the exposure of this 'promiscuous relationship' [77,80]. This provides an example of how political struggles against dams have consequences that extend far beyond the immediate site of construction and contestation. As evident in the multilevel movement against the Narmada dams, the storylines of anti-dam groups can influence wider policies of dam construction. A similar process is evident in the case of Belo Monte—with the process of repoliticization extending beyond the immediate locality of Belo Monte's construction to provide a moment in which such a decision to archive the São Luiz do Tapajós dam was possible.

## 8. Conclusions

Although the social and environmental impacts of large hydroelectric dams have led to widespread criticism of the infrastructure, pro-dam actors continue to legitimize these projects by asserting the benefits associated with their construction. This article has explored one particular storyline of legitimacy, that of the 'national interest', in which the planning and construction of the Belo Monte project has been foregrounded within wider assertions of the economic development and shared benefits. These pro-dam assertions of the 'national interest' function to depoliticize Belo Monte, asserting it as an apolitical and technical project and raising it above the contestation and antagonism of everyday politics. In exploring these storylines, this article has demonstrated how Mouffean conceptions of the political provide a fruitful route to understanding how those proposing water infrastructure projects adopt certain storylines to restrict the potential of public dissent and disagreement. By linking a project to notions of economic development and wider benefits (such as the form of the energy generated), pro-dam actors positioned Belo Monte as a technical pursuit devoid of political character and beyond contention. However, this paper has also demonstrated how such depoliticizing storylines can be reversed, with the repoliticization of projects remaining a possibility for the opposition groups that seek to contest contemporary hydropower projects. In addressing the pro-dam assertions of the 'national interest', anti-dam actors in Brazil have illuminated the political and commercial interests behind the planning and construction of Belo Monte. In doing

so, anti-dam actors reground the project within a wider context of political interests, corruption scandals, and the circumvention of judicial redress. The forwarding of this storyline contests the 'rendering technical' of Belo Monte and reasserts the project's location within the political terrain of agonism. Belo Monte is rendered political, becoming tied to wider political processes and controversies. This recontextualization of the project functions to repoliticize its construction, rendering visible the numerous interests involved and discrediting claims of the project being above everyday politics.

Although Belo Monte will be built, and the Xingu River fragmented, the opposition movement against it demonstrates both how the 'national interest' remains both malleable and contestable and how anti-dam actors are able to alter the trajectory of wider patterns of dam construction. Whilst the political may be concealed in policies of infrastructure construction, it can also be illuminated—with the antagonism of dam construction never fully hidden and able to influence developments beyond its immediate site and impacts.

**Funding:** This research was funded by the Economic and Social Research Council, Grant Number 1325180.

**Acknowledgments:** The author thanks Jessica Hope, Edward Cole, and the two anonymous reviewers for their constructive comments on previous drafts.

**Conflicts of Interest:** The author declares no conflict of interest.

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
