# Peer review of "Disputing the ‘National Interest’: The Depoliticization and Repoliticization of the Belo Monte Dam, Brazil"

_water, doi:10.3390/w11010103_

Round 1

Reviewer 1 Report

This is an interesting article on the Belo Monte dam in Brasil. The article is appropriate for this journal, and is also seems to be in line with the remits of the special issues of which is part, particularly for its analysis of contestations regarding the use of water. Yet, while the article makes some interesting points about the notion of the political, these seem to be still a bit underdeveloped, and I suggest that the author elaborates a bit more on this. Indeed, while the author highlights how the dam has become a political object due to local and national opposition movements, their impact is negligible as the dam has nevertheless been built and is almost complete. As written, the bits in which the author praises these contestations are therefore weak, as the value of these movements (if any) in creating networks of resistance, or in foregrounding counter hegemonic tactics has to be better outlined.

Also, can the author elaborate on how the data was analysed? I suppose discourse analysis was the primary method, and I would welcome a couple of lines explaining how this was carried out

It is also worth mentioning some literature that explored the role of dams/water in relation to the nation state and the political (see for instance Swyngedouw, Menga, Kaika or Allouche), which seems particularly fitting in regard to the bits on national interest. On pro-dam justifiers, the work of Khagram also seems relevant. Furthermore, the author rightly takes Mouffe as a starting point to examine politics and the political, but here the work of Swyngedouw on le politique / la politique and nature society relations needs to also be acknowledged.

I suggest that the author also explains that hydroelectricity is not that clean as dam-proponents claim, as recent studies have highlighted the considerable CO2 emissions caused by seaweed decomposing in reservoirs.

Minor comments

Rather than a hyphen there should be a comma at the end of the 1st line of the abstract

Is there a bit missing in subheading 3 “Constructing a dam…”?

Line 96: I guess the right word is context, not contest

Line 112: Should read “remains”. In general, please carefully proof-read the piece, as there are several typos (see also, for instance “as” in line 122)

Author Response

Dear Reviewer #1,

I thank you for your detailed and constructive comments on the manuscript, ‘Disputing the ‘national interest’: The depoliticization and repoliticization of the Belo Monte dam, Brazil’ submitted to Water on 31 October 2018.

In response to your comments, I have updated the attached manuscript in a number of ways. These different changes are outlined below, structured in response to your recommendations.

Greater efforts have been made to develop discussions of the political further. This includes further reflections on the significance of anti-dam movements in reconfiguring dominant storylines of legitimacy. Drawing on a fuller discussion of the resistance to the Narmada dams in India, it is now argued that the anti-dam movement against Belo Monte alters the trajectory of wider patterns of dam-building – with the process of reconfiguration and repoliticization leading to results that extend beyond the immediate site of the project’s construction.  In making this argument, I assert that, although the impact of the campaign against Belo Monte may be what you define as ‘negligible’, its wider significance can be found in the 2016 removal of the São Luiz do Tapajós project from the environmental licensing process. Anti-dam actors locate both projects within similar storylines - asserting that the ‘archival’ of the São Luiz do Tapajós can be traced to the illumination of links between Belo Monte and the Lava Jato corruption scandal. Whilst I disagree with the notion that the construction of a dam results in the denial of a movement’s significance, the links between the separate fates of these two projects is outlined and emphasised.

In response to your request for a more-expansive discussion of scholarship that details the links between water’s infrastructure and the nation-state, I have re-developed the section on the ‘national interest’ to include a greater reflection on the work of others and the location of hydroelectric projects within the terrain of state-building and economic development (please see: Lines 188-202).

The section detailing the research design and source materials explored has been updated to include a further delineation of how the data was analysed and understood within the research process (please see: Lines 101-109).

I thank you for your recommendation of the work of Swyngedouw on the post-political as a means to improve the theoretical approach adopted. I have drawn on Swyngedouw’s work within the discussion of depoliticisation (please see Lines 263-266). I have also included a footnote highlighting Swyngedouw’s use of the work of Jacques Ranciere to make such an argument (found on p.6). Following this, I understand that Ranciere’s tripartite distinction between policy (what he labels ‘the police’) from both ‘politics’ and ‘the political’ as similar to Mouffe’s division of Mouffe’s division of politics and the political – with the institutions of society (or ‘the police’) separate from the terrain of political debate (‘the political’).

Greater reference has been made to the disputed sustainability of contemporary hydropower projects, with recent research on the environmental impacts of such infrastructure highlighted in a footnote on p.4.

I have responded to your more-minor comments within the manuscript, resulting in a number of edits. This is with the exception of the recommended replacement of the term ‘contest’ with ‘context’ (found on Line 96 in the original manuscript, Line 12 in the submitted updated). With this passage describing a process of contentious interaction, it is important to highlight this confrontation between pro- and anti-dam actors. In addition, greater efforts have been made to proofread and edit the updated manuscript – resulting in the identification and mitigation of a number of smaller mistakes. I thank you for your patience with the previous manuscript.

In addition to these changes, I have made greater efforts to improve the clarity of the manuscript – resulting in the inclusion of further signposting of the central argument, the provision of a paragraph detailing the structure of analysis, a recalibration of the Abstract, and the removal of a number of additional paragraphs that distract from the argument.

Once again, I thank you for your prompt and helpful comments on the attached manuscript. I hope that the updated version submitted represents a fuller, more-effective forwarding of my ideas and argument.

Reviewer 2 Report

The paper addresses a highly relevant topic in water resources management, namely hydropower developmemt and the contestation from anti and pro mega dams. Moreover the structure of the paper is well defined and the data well sopported by the framework of analysis. 

The data and its sources are sufficient, as well as the bibliographic references are relevant. In general the paper is easy to follow. 

Only minor comments are presented in the paper, which can help clarify some aspects.

Author Response

Dear Reviewer #2,

I thank you for your precise and constructive comments on the manuscript, ‘Disputing the ‘national interest’: The depoliticization and repoliticization of the Belo Monte dam, Brazil’ submitted to Water on 31 October 2018.

In response to your comments, I have updated the attached manuscript to represent your required changes. I hope that the updated version submitted represents a fuller, more-effective forwarding of my ideas and argument.

Round 2

Reviewer 1 Report

I am satisfied by these revisions. The paper has now more theoretical depth and is in general stronger. I recommend to accept it for publication

Author Response

Dear Reviewer #1,

I thank you for you comments on the manuscript 'Disputing the 'national interest': The depoliticization and repoliticization of the Belo Monte dam, Brazil'received on 20th December 2018. 

In response to the comments provided, I have conducted a close proofreading of the manuscript, focused on detecting and correcting issues of repetition. I enclose the updated manuscript with thanks. 

Note to Editors: I have included a word document, with track changes included. If you require a cleaner document or pdf copy, please do not hesitate to let me know.
